# Interpregnancy intervals and adverse birth outcomes in high-income countries: An international cohort study

Gizachew A. Tessema[1,2]*, M. Luke Marinovich[1], Siri E. Håberg[3], Mika Gissler[4,5], Jonathan A. Mayo[6], Natasha Nassar[7], Stephen Ball[8], Ana Pilar Betrán[9], Amanuel T. Gebremedhin[1], Nick de Klerk[10], Maria C. Magnus[3,11,12], Cicely Marston[13], Annette K. Regan[1,14], Gary M. Shaw[6], Amy M. Padula[15], Gavin Pereira[1,3]

1 Curtin School of Population Health, Curtin University, Perth, Western Australia, Australia, 2 School of Public Health, University of Adelaide, Adelaide, South Australia, Australia, 3 Centre for Fertility and Health (CeFH), Norwegian Institute of Public Health, Oslo, Norway, 4 Information Services Department, Finnish Institute for Health and Welfare, Helsinki, Finland, 5 Department of Neurobiology, Care Sciences and Society, Karolinska Institute, Stockholm, Sweden, 6 Department of Pediatrics, March of Dimes Prematurity Research Center, Stanford University, Stanford, CA, United States of America, 7 Children's Hospital at Westmead Clinical School, Faculty of Medicine and Health, University of Sydney, New South Wales, Australia, 8 Curtin School of Nursing, Curtin University, Perth, Western Australia, Australia, 9 UNDP/UNFPA/UNICEF/WHO/World Bank Special Programme of Research, Development and Research Training in Human Reproduction, Department of Reproductive Health and Research, World Health Organization, Geneva, Switzerland, 10 Telethon Kids Institute, University of Western Australia, Subiaco, Western Australia, Australia, 11 MRC Integrative Epidemiology Unit at the University of Bristol, Bristol, United Kingdom, 12 Population Health Sciences, Bristol Medical School, Bristol, United Kingdom, 13 Faculty of Public Health and Policy, London School of Hygiene and Tropical Medicine, London, United Kingdom, 14 School of Public Health,Texas A&M University, College Station, Texas, United States of America, 15 Department of Obstetrics, Gynecology and Reproductive Sciences, University of California, San Francisco, CA, United States of America

* gizachew.tessema@curtin.edu.au

**Data Availability Statement:** The data underlying this study are not publicly available due to agreements that restrict our ability to share the unit

## Abstract

### Background

Most evidence for interpregnancy interval (IPI) and adverse birth outcomes come from studies that are prone to incomplete control for confounders that vary between women. Comparing pregnancies to the same women can address this issue.

### Methods

We conducted an international longitudinal cohort study of 5,521,211 births to 3,849,193 women from Australia (1980–2016), Finland (1987–2017), Norway (1980–2016) and the United States (California) (1991–2012). IPI was calculated based on the time difference between two dates—the date of birth of the first pregnancy and the date of conception of the next (index) pregnancy. We estimated associations between IPI and preterm birth (PTB), spontaneous PTB, and small-for-gestational age births (SGA) using logistic regression (between-women analyses). We also used conditional logistic regression comparing IPIs and birth outcomes in the same women (within-women analyses). Random effects meta-analysis was used to calculate pooled adjusted odds ratios (aOR).

record data on ethical and legal grounds. The data are sensitive, and de-identified but potentially re-identifiable. Requests to access these data can be submitted to the following entities: New South Wales, Australia - NSW Centre for Health Record Linkage website: https://www.cherel.org.au/. Western Australia - Data Linkage Western Australia website: https://www.datalinkage-wa.org.au/. Norway, websites: https://helsedata.no/. Finland: Social and Health Data Permit Authority Findata website: https://findata.fi/en/.

**Funding:** This work was supported by funding from the National Health and Medical Research Council, including an Early Career Fellowship [#GNT1138425 to AR], investigator grant [#APP1195716 for GAT, #APP1173991 for GP], Career Development Fellowship [#GNT1067066 to NN] and project grants [#GNT1099655 to GP, NN and SJB; # GNT1141510 to AKR and GP, # GNT1173991 to GP]. MCM works at the MRC Integrative Epidemiology Unit at the University of Bristol which receives infrastructure funding from UK Medical Research Council (MC/UU/12013/5), and she is also supported by the following fellowship from the UK Medical Research Council (MR/M009351/1). This work was partly supported by the Research Council of Norway through its Centres of Excellence funding scheme [# 262700 to SEH and GP]. This work was partly supported by the March of Dimes Prematurity Research Center at Stanford University School of Medicine and by the National Institutes of Health (R03HD090243). The funders have no role in the design, analysis and interpretation of the results in the study.

**Competing interests:** The authors have declared that no competing interests exist.

## Results

Compared to an IPI of 18–23 months, there was insufficient evidence for an association between IPI <6 months and overall PTB (aOR 1.08, 95% CI 0.99–1.18) and SGA (aOR 0.99, 95% CI 0.81–1.19), but increased odds of spontaneous PTB (aOR 1.38, 95% CI 1.21–1.57) in the within-women analysis. We observed elevated odds of all birth outcomes associated with IPI ≥60 months. In comparison, between-women analyses showed elevated odds of adverse birth outcomes for <12 month and >24 month IPIs.

## Conclusions

We found consistently elevated odds of adverse birth outcomes following long IPIs. IPI shorter than 6 months were associated with elevated risk of spontaneous PTB, but there was insufficient evidence for increased risk of other adverse birth outcomes. Current recommendations of waiting at least 24 months to conceive after a previous pregnancy, may be unnecessarily long in high-income countries.

## Introduction

Both short and long interpregnancy intervals (IPI) have been associated with increased risk of adverse birth outcomes such as preterm birth (PTB, < 37 weeks of gestation) [1–6], small-for-gestational age (SGA) [7–9] and term low birth weight (LBW) [4, 10, 11], among other adverse perinatal outcomes [12–16]. Meta-analyses have reported that short (<6 months) and long (≥60 months) IPIs are associated with increased risk of PTB, SGA, term LBW compared to an IPI of 18–23 months [17]. In 2005, the World Health Organization recommended waiting at least 24 months before attempting pregnancy following a live birth [18]. This was based on limited evidence and analyses based on studies available at the time, mostly from low- and middle-income countries.

Although adverse effects of IPIs have been consistently reported for between-women (unmatched) cohort studies, results from more recent within-women (matched) studies that compare different pregnancies from the same women suggest bias in the between-women studies, due to unmeasured or inadequately controlled maternal characteristics that remain stable between pregnancies [19–22]. The first study of this type was conducted by Ball *et al.* [20] in Western Australia and reported that the increased risk of PTB, term LBW and SGA after a short IPI was lower when comparing subsequent pregnancies to the same women. Swedish [22] and Californian [23] studies have since reported substantially smaller associations between short intervals and adverse birth outcomes when using such a within-women design [22, 23]. A Canadian study comparing outcomes of pregnancies within the same women reported that short IPIs were associated with reduced, not increased odds of PTB [21]. A systematic review also indicated that results for the associations between IPI and adverse births outcomes were inconsistent [24]. Therefore, the direction and magnitude of the association between short intervals and adverse birth outcomes remains unclear.

We conducted an international longitudinal cohort study to investigate the association between IPI and adverse birth outcomes in four high-income countries, using within-women and between-women analyses.

## Methods

### Study design and population

We conducted an international longitudinal cohort study on the association between IPI and birth outcomes using individual-level perinatal records from four high-income countries: Australia (Western Australia [WA] and New South Wales [NSW]) (1980–2016), Finland (1987–2017), Norway (1980–2016), and the United States (California) (1991–2012). Descriptions of the perinatal data sources are described in the published protocol [25]. In summary, perinatal records of birth with complete or near-complete coverage were obtained from the Midwives Notifications System (WA), the Perinatal Data Collection (NSW), the Medical Birth Register of Finland, and the Medical Birth Register of Norway. For California, we obtained a linked birth cohort file that merged fetal death, birth, and infant death certificates provided by the Office of Statewide Health Planning and Development (OSHPD) [25]. These data sources contain unique identifiers for women and births that allow linkage of births to their mothers. We followed the Strengthening the Reporting of Observational Studies in Epidemiology (STROBE) checklist to report our findings [26] (S1 Checklist).

### Inclusion criteria

There were 11,383,557 births from the participating countries/states during the study period, 1980 to 2017 (Australia: 3,074,232; Finland: 1,850,446; Norway: 2,094,171; California: 4,364,708). We excluded birth records for multiple gestations, with gestational age <22 or >44 weeks, with missing birth weight or weight <500g, with unspecified sex, and records with missing maternal age or age <14 years at birth [25]. These exclusions resulted in 9,902,167 births, and of these we excluded records without prior registered births (i.e. no IPI), leaving 5,521,206 births to 3,849,191 women for the between-women analyses (S1 Fig). Finally, we identified births to women with at least two or more IPIs resulting in a cohort of 2,905,703 births to 1,233,688 women for within-women analyses. Study entry was defined for all women as the earliest birth during the study period with an IPI.

### Outcomes

We investigated three birth outcomes: PTB, spontaneous PTB and SGA. PTB was defined as birth <37 completed weeks of gestation. For most countries, spontaneous PTB was defined as PTB with spontaneous onset of labour, while for California, spontaneous PTB was defined as a PTB with any of the following three criteria: premature rupture of membranes, premature labour or tocolysis. SGA was defined as a birthweight in the lowest 10th centile based on the national birthweight distributions in each country by final week of gestational age and sex.

### Exposure

IPI was the main exposure and calculated based on the time difference between two dates—the date of birth of the first pregnancy and the date of conception of the next (index) pregnancy (birth date minus gestational length) [27]. In all countries, ultrasound was routinely used to confirm gestational age and where unavailable it was calculated based on the date of the menstrual period. IPI was categorised as: 0–5, 6–11, 12–17, 18–23, 24–59, 60–119, and ≥120 months, with 18–23 months used as the reference category [20].

### Statistical analyses

All analyses were adjusted using a prognostic score derived with logistic regression of the birth outcome regressed on time-varying covariates. Prognostic score adjustment [28, 29] was

adopted to minimise the within-women collinearity between the exposure and the time-vary-ing adjustment variables. The variables included when creating prognostic score were mater-nal age (<20, 20–24, 25–29, 30–34, 35–39, and ≥40 years), parity (1, 2, 3, ≥4) and year of birth (5-year categories). We also assessed sensitivity to (i) inclusion of stillbirths in the IPI defini-tions [30, 31]; (ii) between-women analyses for women with at least two IPIs (at least three births); and (iii) additional adjustment for socioeconomic status (SES) for the cohorts with this information available: Australia, Finland, and California. For Australia, SES was derived based on scores from the Index of Relative Socio-Economic Disadvantage (quintiles), a composite of education, skilled occupation status, and household income [32]. For Finland, SES was recorded using maternal occupational status at the time of birth and included four categories (blue collar, lower-white collar, upper-white collar workers, and 'other'). For California, maternal educational attainment was considered as a proxy measure of SES and comprised four categories (some high school or less, high school diploma/equivalent, some college, col-lege graduate or more). SES adjustment was not possible for Norway for which no SES infor-mation was available. Furthermore, we conducted additional analyses to assess sensitivity of our results for the inclusion of births of from higher-order parity women by restricting our cohort to the first three births (parity 0, 1 and 2) in Australia and Norway and for the inclusion of births with gestational ages estimated using LMP by restricting births from 1990 onwards in Norway when ultrasound is being widely used to estimate gestational ages.

Conventional logistic regression was used for between-women analyses and conditional logistic regression applied for within-women analyses that compared the outcomes and expo-sures of subsequent pregnancies to the same women. First, analyses were conducted for each country and each outcome measure separately. Second, country-specific adjusted odds ratios (aOR) for the between-women and within-women analyses were pooled using the inverse vari-ance method with random intercepts for country [33]. Heterogeneity in the estimates between countries was identified with the $I^2$ statistic [34]. Analyses were done with SAS version 9.4 and Stata version 14.

### Ethical statement

Ethics approval was obtained from Human Research Ethics Committees and Institutional Review Boards from the Departments of Health in Western Australia and New South Wales, Curtin Uni-versity, Stanford University, and the Norwegian Regional Committees for Medical and Health Research Ethics of South/East Norway. Each committee provided a waiver of consent for use of par-ticipants' data. For Finland, ethical approval was not required for studies based on registry data.

## Results

### Cohort characteristics at study entry

Approximately 89–93% of women entered the study cohort at parity 1 in Australia, Finland, Norway, and 98% in California for the within-women study (Table 1). For the between-women study, except in California, 87–90% of women entered the study cohort at parity 1 (S1 Table). In the within-women study, at study entry, the highest proportion of women were between the ages of 25 and 29 years in Australia (36%), Finland (43%), and Norway (45%); but were between 20 and 24 years in California (35%).

### Country specific analysis

We observed that 6.7% of all births had IPI <6 months, ranging from 3.8% of births in Norway to 7.9% in California (Table 2). The proportion of births after an IPI of 18–23 months was

**Table 1. Maternal characteristics at study entry (first birth in the included cohort)* for the within-women analysis (n = 1,233,688).**

| Characteristics | Australia | Finland | Norway | California |
|---|---|---|---|---|
| | N (%) | N (%) | N (%) | N (%) |
| **Total women** | 306,639 | 202,252 | 219,770 | 505,027 |
| **Parity**** | | | | |
| 1 | 273,911 (89.3) | 186,615 (92.3) | 205,117 (93.4) | 495,390 (98.1) |
| 2 | 20,649 (6.7) | 9,352 (4.6) | 10,851 (4.9) | 9,222 (1.8) |
| ≥3 | 12,079 (3.9) | 6,285 (3.10) | 3,802 (1.7) | 415 (0.1) |
| **Maternal age (years)** | | | | |
| <20 | 12,764 (4.2) | 2,749 (1.4) | 2,194 (1.0) | 50,034 (9.9) |
| 20–24 | 75,219 (24.5) | 48,734 (24.1) | 51,674 (23.5) | 177,492 (35.2) |
| 25–29 | 109,210 (35.6) | 86,098 (42.6) | 98,248 (44.7) | 148,185 (20.3) |
| 30–34 | 85,271 (27.8) | 52,316 (25.9) | 56,904 (25.9) | 100,803 (20.0) |
| 35–39 | 22,686 (7.4) | 11,442 (5.7) | 10,215 (4.65) | 26,882 (5.3) |
| ≥40 | 1,489 (0.5) | 913 (0.5) | 535 (0.2) | 1,631 (0.3) |
| **Birth year** | | | | |
| 1980–1989 | 29,804 (9.7) | 9,400 (4.6) | 54,220 (24.7) | - |
| 1990–1999 | 79,256 (25.8) | 87,752 (43.4) | 79,037 (36.0) | 199,063 (30.5) |
| 2000–2009 | 147,576 (48.1) | 75,221 (37.2) | 70,198 (31.9) | 290,640 (57.5) |
| 2010 or later | 50,003 (16.3) | 29,879 (14.8) | 16,315 (7.4) | 15,324 (3.0) |
| **Socioeconomic index for areas (quintile)** | | | | |
| Lowest | 71,259 (23.2) | N/A | N/A | N/A |
| Low | 62,479 (20.4) | N/A | N/A | N/A |
| Middle | 56,814 (18.5) | N/A | N/A | N/A |
| High | 52,143 (17.0) | N/A | N/A | N/A |
| Highest | 47,595 (15.5) | N/A | N/A | N/A |
| Other/unknown/missing | 16,349 (5.3) | N/A | N/A | N/A |
| **Maternal occupation** | | | | |
| Upper-white collar worker | N/A | 22,675 (13.9) | N/A | N/A |
| Lower-white collar worker | N/A | 74,147 (36.7) | N/A | N/A |
| Blue collar worker | N/A | 32,702 (16.2) | N/A | N/A |
| Other or unknown | N/A | 67,728 (33.5) | N/A | N/A |
| **Highest level of maternal education** | | | | |
| Some high school or less | N/A | N/A | N/A | 99,696 (19.7) |
| High school diploma/Equivalent | N/A | N/A | N/A | 169,195 (33.5) |
| Some college | N/A | N/A | N/A | 120,038 (23.8) |
| College graduate or more | N/A | N/A | N/A | 109,357 (21.7) |
| Missing | N/A | N/A | N/A | 6,741 (1.3) |
| **Number of IPIs** | | | | |
| 2 | 220,175 (71.8) | 144,141 (71.3) | 177,636 (80.8) | 385,449 (76.3) |
| ≥3 | 86,464 (28.2) | 58,111 (28.7) | 42,134 (19.2) | 119,578 (23.7) |

*Study entry is defined as the earliest birth (second birth) at which the women appeared within the included cohort during the study period.

**Women parity at the birth when entering to the cohort (i.e the second birth after interval). IPI -_Interpregnancy intervals. N/A—data were not available in these countries.

similar across countries ranging from 12.9% (California) to 14.6% (Australia). The prevalence of PTB ranging from 3.9% (Finland) to 8.6% (California); spontaneous PTB was between 2.9% and 4.0% for all countries; and SGA was similar for all countries ranging from 6.7% (Norway) to 8.0% (Australia) (S2 Table).

**Table 2. Interpregnancy interval distribution for births in within-women analyses across the four countries (n = 2,905,703).**

| IPI category (months) | Australia N (%) | Finland N (%) | Norway N (%) | California N (%) | Total N (%) |
|---|---|---|---|---|---|
| **<6** | 54,847 (7.4) | 30495 (5.6) | 18,617 (3.8) | 91,473 (7.9) | 195,432 (6.7) |
| **6–11** | 132,930 (18.0) | 94,415 (18.6) | 58,987 (12.0) | 172,445 (14.8) | 458,778 (15.8) |
| **12–17** | 144,015 (19.5) | 94,069 (18.5) | 75,168 (15.2) | 185,782 (16.0) | 499,035 (17.2) |
| **18–23** | 107,880 (14.6) | 66,998 (13.2) | 66,934 (13.6) | 150,103 (12.9) | 391,916 (13.5) |
| **24–59** | 230,998 (31.3) | 159685 (31.4) | 194,292 (39.4) | 414,553 (35.6) | 999,529 (34.3) |
| **60–119** | 60,090 (8.1) | 54622 (10.7) | 67,966 (13.8) | 134,038 (11.5) | 316,723 (10.9) |
| **≥120** | 8,337 (1.1) | 8749 (1.7) | 11,157 (2.3) | 16,058 (1.4) | 44,301 (1.5) |

*Numbers in brackets are column percentages for IPI categories in each country

The between-women analyses showed elevated odds of PTB ranging from 1.42 (California) to 2.06 (Norway); spontaneous PTB ranging from 1.67 (California) to 2.04 (Norway); and SGA ranging from 1.07 (Finland) to 1.42 (California) for short IPIs of <6 months compared to IPI of 18–23 months in all countries. For long IPIs, the odds of PTB, sPTB, and SGA from between-women analyses were elevated for all countries. In the within-women analyses, compared to an IPI of 18–23 months, there were elevated odds of PTB after IPIs of <6 months and 6–11 months in Australia and California, but no association for Finland and Norway. The odds of spontaneous PTB after an IPI <6 months were elevated for all countries, ranging from 1.21 (Norway) to 1.66 (Australia). Except for California, the odds of SGA were reduced in Australia, Finland and Norway after an IPI of <6 months compared with an IPI of 18–23 months. For long IPIs, the odds of PTB, sPTB, and SGA from within-women analyses were elevated for all countries (Table 3).

## Pooled analyses

Pooled adjusted between-women analyses indicated association between IPI and adverse birth outcomes. Elevated odds of all adverse outcomes were observed for short IPIs (<6 months and 6–11 months) and long IPI categories (24–59 months, 60–119 months, and ≥120 months) (S2–S4 Figs).

Pooled adjusted within-women analyses indicated no or weak association between IPIs of <6 months (aOR 1.08, 95% CI: 0.99, 1.18) and PTB, compared with 18–23 months of IPI (Fig 1). Odds of spontaneous PTB were elevated for IPIs <6 months (aOR = 1.38, 95% CI: 1.21, 1.57) (Fig 2). For SGA, there was no observed association for IPIs of <6 months (aOR = 0.99, 95% CI: 0.81, 1.19) (Fig 3). The pooled estimate for long IPI categories of 60–119 months and ≥120 months were consistently associated with elevated odds of all adverse birth outcomes (Figs 1–3).

For between-women and within-women analyses, the direction of association between IPI and the adverse birth outcomes was mostly consistent between countries, except for IPI <6 months for PTB and SGA in the within-women analyses (Figs 1 and 3). The magnitude of associations varied across countries for both between-women and within-women analyses. Heterogeneity between countries ($I^2$) generally exceeded 50% for all outcomes (Figs 1–3, S2–S4 Figs).

## Sensitivity analyses

Additional adjustment for SES resulted in negligible change to the results (S3 Table). Restriction to live births, also had negligible influence on the observed associations for most countries

**Table 3. Adjusted odds ratios (aOR) and 95% confidence intervals (CI) between interpregnancy interval and adverse birth outcomes among births in between-women\* and within-women\*\* analyses across the four countries.**

| Outcome by country | Interpregnancy interval | | | | | | |
|---|---|---|---|---|---|---|---|
| | <6 months | 6–11 months | 12–17 months | 18–23 months | 24–59 months | 60–119 months | ≥120 months |
| **PTB** | aOR (95% CI) | | | | | | |
| **Australia** | | | | | | | |
| Between-women | 1.68 (1.63, 1.73) | 1.13 (1.10, 1.16) | 0.99 (0.97, 1.02) | Ref | 1.16 (1.13, 1.19) | 1.59 (1.55, 1.64) | 2.14 (2.03, 2.27) |
| Within-women | 1.22 (1.15, 1.29) | 1.08 (1.03, 1.14) | 1.00 (0.95, 1.05) | Ref | 1.06 (1.02, 1.11) | 1.35 (1.27, 1.43) | 1.71 (1.52, 1.92) |
| **Finland** | | | | | | | |
| Between-women | 1.62 (1.53, 1.70) | 1.06 (1.01, 1.10) | 0.99 (0.95, 1.10) | Ref | 1.12 (1.08, 1.16) | 1.41 (1.35, 1.47) | 1.73 (1.61, 1.86) |
| Within-women | 1.00 (0.91, 1.09) | 1.02 (0.95, 1.10) | 0.95 (0.88, 1.02) | Ref | 1.00 (0.94, 1.07) | 1.26 (1.16, 1.37) | 1.67 (1.44, 1.95) |
| **Norway** | | | | | | | |
| Between-women | 2.06 (1.96, 2.17) | 1.17 (1.12, 1.22) | 1.02 (0.98, 1.06) | Ref | 1.11 (1.08, 1.15) | 1.51 (1.45, 1.57) | 1.91 (1.82, 2.02) |
| Within-women | 0.97 (0.88, 1.06) | 1.01 (0.93, 1.09) | 1.03 (0.96, 1.12) | Ref | 1.08 (1.02, 1.15) | 1.38 (1.29, 1.49) | 1.71 (1.51, 1.93) |
| **California** | | | | | | | |
| Between-women | 1.42 (1.39, 1.45) | 1.16 (1.14, 1.18) | 1.06 (1.05, 1.08) | Ref | 1.12 (1.10, 1.14) | 1.41 (1.39, 1.44) | 1.82 (1.76, 1.88) |
| Within-women | 1.13 (1.09, 1.18) | 1.10 (1.06,1.14) | 1.07 (1.03,1.11) | Ref | 1.00(0.97, 1.03) | 1.17 (1.13, 1.22) | 1.42 (1.31, 1.54) |
| **Spontaneous PTB** | | | | | | | |
| **Australia** | | | | | | | |
| Between-women | 1.93 (1.85, 2.01) | 1.22 (1.18, 1.27) | 1.03 (0.99, 1.07) | Ref | 1.13 (1.10, 1.17) | 1.56 (1.50, 1.63) | 1.99 (1.84, 2.16) |
| Within-women | 1.66 (1.54, 1.79) | 1.30 (1.22, 1.39) | 1.07 (1.00, 1.14) | Ref | 0.96 (0.90, 1.02) | 1.10 (1.02, 1.19) | 1.28 (1.09, 1.51) |
| **Finland** | | | | | | | |
| Between-women | 1.77 (1.66, 1.88) | 1.14 (1.09, 1.20) | 1.04 (0.99, 1.09) | Ref | 1.15 (1.10, 1.20) | 1.45 (1.37, 1.52) | 1.76 (1.62, 1.92) |
| Within-women | 1.32 (1.19, 1.46) | 1.22 (1.12, 1.34) | 1.04 (0.96, 1.14) | Ref | 1.05 (0.96, 1.13) | 1.35 (1.22, 1.49) | 1.77 (1.47, 2.13) |
| **Norway** | | | | | | | |
| Between-women | 2.04 (1.92, 2.18) | 1.22 (1.16, 1.28) | 1.04 (0.99, 1.09) | Ref | 1.06 (1.02, 1.10) | 1.40 (1.34, 1.47) | 1.78 (1.64, 1.90) |
| Within-women | 1.21 (1.08, 1.36) | 1.10 (1.00, 1.21) | 1.04 (0.95, 1.14) | Ref | 0.93 (0.86, 1.01) | 1.12 (1.02, 1.23) | 1.29 (1.09, 1.51) |
| **California** | | | | | | | |
| Between-women | 1.67 (1.62, 1.72) | 1.15 (1.12, 1.18) | 1.03 (1.00, 1.06) | Ref | 1.15 (1.12, 1.17) | 1.52 (1.48, 1.56) | 1.96 (1.88, 2.05) |
| Within-women | 1.34 (1.26, 1.42) | 1.12 (1.06, 1.18) | 1.06 (1.00, 1.13) | Ref | 1.05 (1.00, 1.10) | 1.35 (1.27, 1.43) | 1.65 (1.47, 1.86) |
| **SGA** | | | | | | | |
| **Australia** | | | | | | | |
| Between-women | 1.10 (1.07, 1.13) | 0.99 (0.97, 1.01) | 0.98 (0.96,1.01) | Ref | 1.18 (1.15, 1.20) | 1.65 (1.61, 1.69) | 2.08 (1.98, 2.19) |
| Within-women | 0.94 (0.87, 0.99) | 0.99 (0.95, 1.04) | 1.03 (0.99,1.08) | Ref | 1.04 (1.00, 1.09) | 1.24 (1.17, 1.31) | 1.58 (1.40, 1.78) |
| **Finland** | | | | | | | |
| Between-women | 1.07 (1.03, 1.12) | 0.95 (0.92, 0.98) | 0.98 (0.95, 1.01) | Ref | 1.09 (1.06, 1.12) | 1.35 (1.31, 1.40) | 1.72 (1.63, 1.81) |
| Within-women | 0.89 (0.82, 0.96) | 0.99 (93, 1.05) | 1.00 (0.95, 1.07) | Ref | 1.08 (1.02, 1.13) | 1.24 (1.16, 1.32) | 1.80 (1.59, 2.03) |
| **Norway** | | | | | | | |
| Between-women | 1.28 (1.22, 1.35) | 1.08 (1.05, 1.12) | 1.02 (0.99, 1.05) | Ref | 1.11 (1.08, 1.14) | 1.42 (1.37, 1.46) | 1.83 (1.75, 1.92) |
| Within-women | 0.88 (0.81, 0.97) | 0.98 (0.91, 1.05) | 0.97 (0.91, 1.03) | Ref | 1.05 (1.00, 1.11) | 1.26 (1.18, 1.34) | 1.63 (1.45, 1.82) |
| **California** | | | | | | | |
| Between-women | 1.42 (1.39, 1.46) | 1.08 (1.06, 1.10) | 1.00 (0.98, 1.02) | Ref | 1.13 (1.11, 1.15) | 1.37 (1.34, 1.40) | 1.64 (1.59, 1.70) |
| Within-women | 1.26 (1.20, 1.31) | 1.05 (1.01, 1.09) | 1.00 (0.96, 1.04) | Ref | 1.03 (1.00, 1.07) | 1.17 (1.12, 1.22) | 1.52 (1.39, 1.67) |

aOR—adjusted odds ratio. CI—confidence interval. IPI—interpregnancy interval. PTB—preterm birth. SGA—small for gestational age.

\*Odds ratios calculated using between-women analyses for women with ≥2 births/ ≥1 IPI after prognostic score adjustment for maternal age, parity, and year of birth.

\*\*Odds ratios calculated using within-women analyses for women with ≥3 births/ ≥2 IPIs after prognostic score adjustment for maternal age, parity, and year of birth

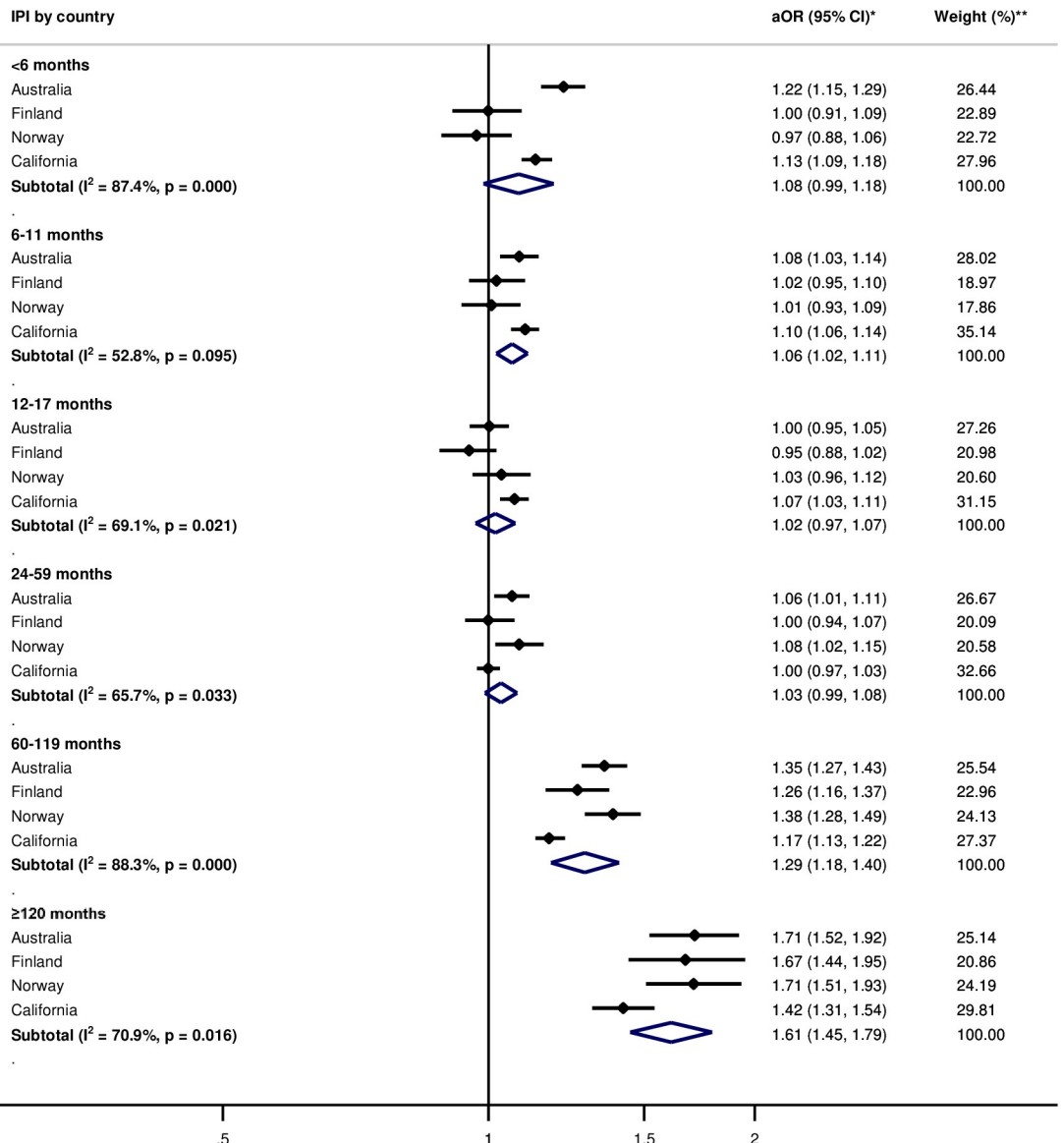

**Fig 1. Adjusted odds ratios of within-women analysis for the association between interpregnancy interval and preterm birth as compared to 18–23 months interpregnancy interval by country.** IPI—Interpregnancy interval. *Adjusted Odds ratios (aOR) and corresponding 95% confidence intervals, adjusted for maternal age, parity, and birth year; the reference IPI category is 18–23 months. **Weights are derived from inverse-variance.

(S4 Table), and an increase in the odds of PTB in Norway (0.97 to 1.32) for IPI <6 months. The results of the between-women analyses restricted to women with at least two IPIs (at least three births) were consistently similar with the results of between-women analyses for women with at least one IPI (S5 Table). Moreover, our additional analysis restricting our cohorts in Australia and Norway to the first three births (parity 0, 1, and 2) for accounting the inclusion of births from women in the higher-order parity did not significantly change the results (S6 Table). Our analysis restricting births from 1990 onwards in Norway, when ultrasound has been widely used to precisely estimate gestational ages, had negligible impact on our results (S7 Table).

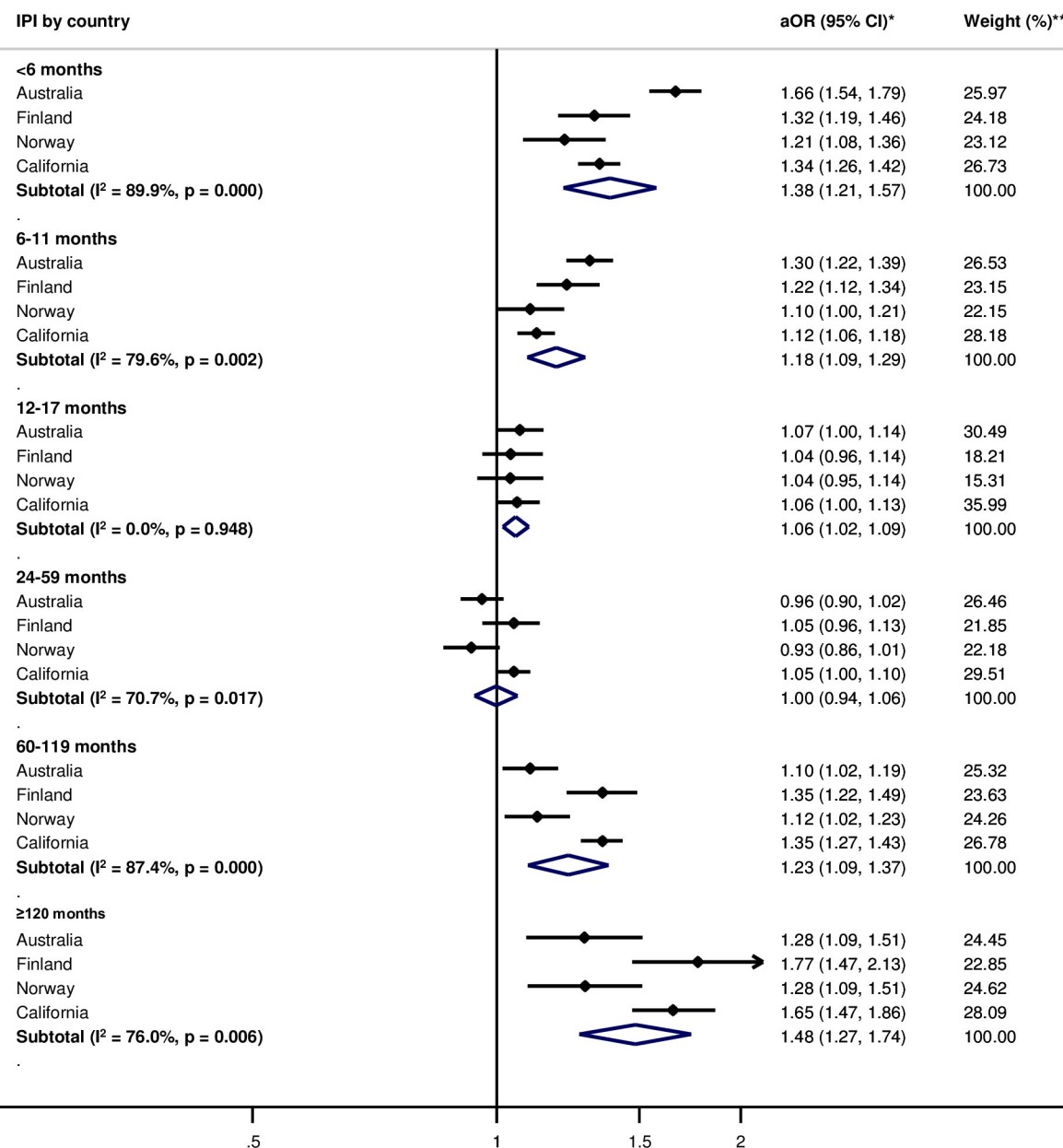

**Fig 2. Adjusted odds ratios for within-women analysis for the association between interpregnancy interval and spontaneous preterm birth as compared to 18–23 months of interpregnancy interval by country.** IPI—interpregnancy interval. *Adjusted Odds ratios (aOR) and corresponding 95% confidence intervals, adjusted for maternal age, parity, and birth year; the reference IPI category is 18–23 months. **Weights are derived from inverse-variance.

## Discussion

This international cohort study of nearly five and half million births indicated associations between both short and long IPIs and adverse birth outcomes in four high-income countries, when pregnancies to different women were compared. After comparing pregnancy outcomes after IPIs within the same women with at least three pregnancies, we observed smaller odds ratios, but consistently elevated odds of all outcomes (PTB, spontaneous PTB, and SGA) for long IPIs. Associations with short IPIs were limited to the spontaneous PTB outcome.

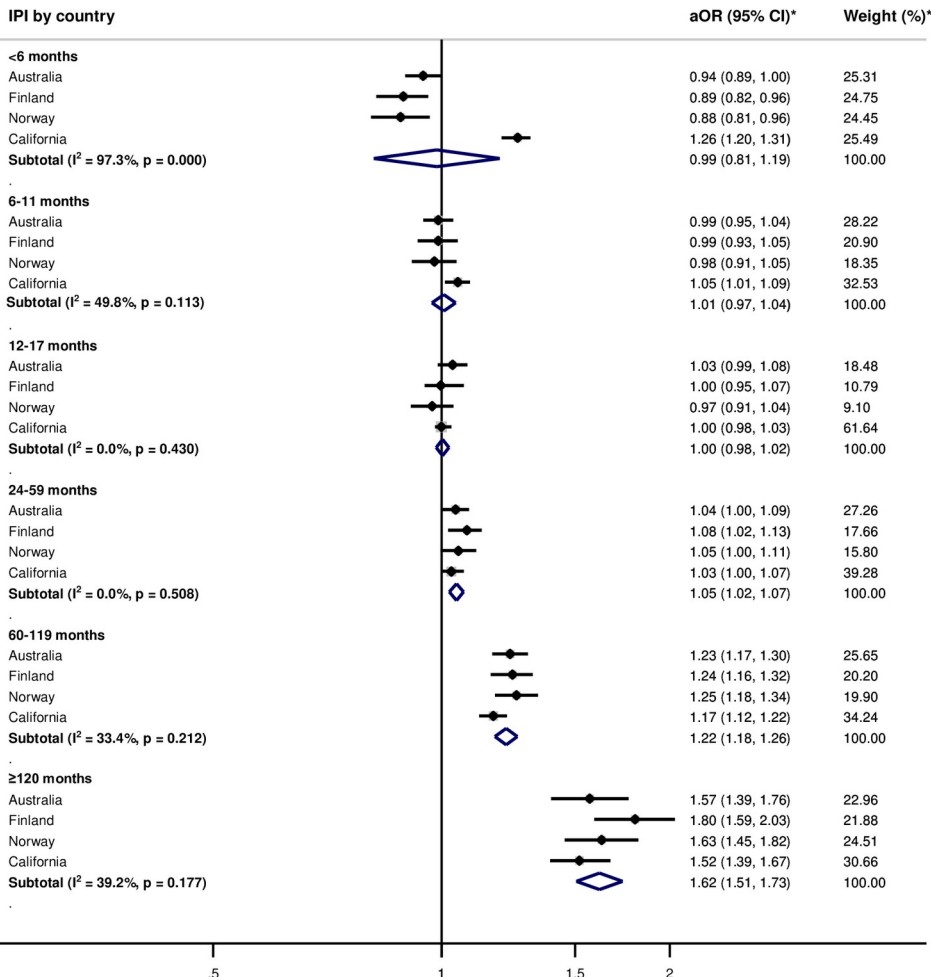

**Fig 3. Adjusted odds ratios for the within-women analysis for the association between interpregnancy interval and small-for-gestational age birth as compared to 18–23 months of interpregnancy interval by country.** IPI—interpregnancy interval. *Adjusted Odds ratios (aOR) and corresponding 95% confidence intervals, adjusted for maternal age, parity, and birth year; the reference IPI category is 18–23 months. **Weights are derived from inverse-variance.

Consistent with previous literature, results from our between-women analyses indicated elevated odds of adverse birth outcomes for both short and long IPIs [1, 8, 10, 17, 35]. Estimates were smaller in within-women analyses, indicating bias away from the null attributed from confounders that vary between-women.

Associations between IPI and adverse birth outcomes have been investigated by comparing pregnancies to the same women in two independent cohorts from Canada and Sweden [21, 22]. Our finding of a lack of association between short IPIs and SGA and PTB was consistent with the results from the Canadian study [21]. However, the Canadian study reported a reduced odds ratio (0.85) for PTB. The Swedish study reported a slightly elevated odds ratio (1.23) of PTB for short IPI [22]. In our study, evidence of associations were less apparent for overall PTB than for spontaneous PTB. Medical intervention shortens the pregnancy length [36] and it is possible that IPI is not as strongly associated with the clinical indications for PTB as it is with spontaneous onset of preterm labour. This hypothesis is supported by results from other between-women studies in Canada and California [1, 23], that found greater odds of a

spontaneous PTB (2.41 for Canada, 1.81 for California) than the odds for medically indicated PTB (1.88 for Canada, 1.35 for California).

We observed elevated odds of adverse birth outcomes for long IPIs for our cohort. While the study from Sweden showed similar results with long IPIs [22], the Canadian study did not [21]. Despite consistent evidence there are currently no recommendations to limit long IPI.

The explanation for the association between IPI and adverse birth outcomes is not well-understood. Given the lack of observed associations between overall PTB and SGA with short IPI, our findings do not provide support for the maternal depletion hypothesis (nutritional depletion) that has been previously suggested, at least in low- and middle-income countries [11, 37, 38]. A potential mechanism for the association between short IPI and spontaneous PTB may be the failure of uterine contraction associated chemicals/proteins, such as *G protein coupled receptors*, to return back to pre-pregnancy levels, which could predispose women of short IPI for preterm onset of labour [3, 39]. Short IPIs also do not permit adequate time to recover from infections that cause inflammatory disease such as endometritis [40], and leave less time for the uterine incision to heal after caesarean section.[37, 41] Short IPIs may leave women with insufficient time to lose excess weight gained during pregnancy. Maternal over-weight or obesity may result in increased risk of inflammatory up-regulation [42], and increased levels of inflammatory proteins (cytokines) may lead to cervical ripening and cause weakening of the membranes and preterm myometrial contractions through prostaglandin activation [43]. Unfortunately, owing to the administrative nature of study data these and other potential mechanistic hypotheses could not be investigated.

Our observation of increased risk of adverse birth outcomes after long IPIs might be explained by the potential gradual decline in uterine adaptation over time, such as changes to uterine blood flow that can result in regression back to a primigravida state [10]. A previous study observed that outcomes of births conceived after a long IPI are similar to perinatal outcomes of births born to women who are pregnant for the first time [10]. In addition, the increased risk of adverse birth outcomes after a long IPI may be due to underlying causes such as sexually transmitted infections or maternal chronic illnesses that lead to both secondary sub-fertility and adverse perinatal outcomes [35, 44]. While partner change has previously been suggested as a potential factor explaining the association between IPI and adverse pregnancy outcomes [45], a recent study in Western Australia indicated that partner changes between pregnancies did not influence the association between IPI and risk of preeclampsia [46].

We could not analytically explore a range of time-varying potential covariates such as pre-pregnancy body mass index, diet during pregnancy, maternal medical conditions and fertility treatment due to lack of complete information in the cohorts across countries. Notably, women who conceive quickly after childbirth might be relatively healthy and relatively more fertile than women who conceive later and therefore less prone to adverse birth outcomes [30]. Fertility treatment is associated with adverse birth outcomes [47, 48] and longer IPIs due to the requirement of a longer period of time to conception in sub-fertile couples [49]. Although we expect that only small proportion of women (1–4%) received fertility treatment in the cohort periods of this study [50–52], the uptake of reproductive technology is rapidly increasing in some countries [53] which may suggest that future studies need to account for assisted reproductive technology. As the primary design for this study was within-women (matched) analysis, conditioning on the outcome (i.e adverse birth outcomes) prior to the interval would introduce potential bias with estimated effect sizes because for those women with two intervals, the outcome of the second pregnancy would also be an effect modifier of the effect of IPI on the third pregnancy. However, a previous analysis using between-women analysis suggested that previous PTB may modify the effect of IPI on PTB in the subsequent pregnancies [54].

Our study also did not adjust for maternal ethnicity/race due to lack of complete information across cohorts and it is possible that the between-women results are prone to confounding by these variables. Since maternal ethnicity/race would not change between pregnancies, our within-women analysis will account this time-invariant confounder and therefore not adjusting for maternal ethnicity/race would not significantly change within-women results. Nonetheless, generalisability to other study populations might still be affected by within-country heterogeneity in race/ethnicity. However, a Californian study reported that the effect of short IPI on the risks of PTB did not substantially differ between non-Hispanic White and non-Hispanic Black women [2]. Given the within women analysis included small number of women with at least two IPIs (three or more births), these women may represent highly selected cohort of women and hence may not be generalisable to the general population. However, the results of the between-women analyses comprising these women were consistent with the results from the between-women analyses comprising at least one IPI (two or more births) over the study period suggesting it is unlikely that our results are necessarily reflective of differences in the two cohorts of women.

Undetected pregnancy loss (miscarriage, induced abortion and stillbirth) inflates IPI and is a marker for adverse birth outcomes in subsequent pregnancies [55]. Unlike previous studies that restricted analyses to consecutive live births [20, 22, 56], we included both live births and stillbirths, but could not account for pregnancy loss before 22 weeks. The associations between IPI and preterm birth might differ by presentation based on timing of delivery (extreme PTB (<28 weeks), very PTB (28–31 weeks), moderate PTB (32–37 weeks)) [3], however, the scope of this study was limited to separately investigate the association between IPI and PTB classified by clinical presentation (spontaneous and overall PTB).

Although there was substantial heterogeneity in the magnitude of the effects, the direction of associations was generally consistent. Part of the heterogeneity in the magnitude of results could be due to the difference in the characteristics of the population. For example, unlike Australia and California, the population in Norway and Finland represent relatively ethnically homogenous populations. An alternative explanation for the large observed heterogeneity ($I^2$) is that most of the results were similar between countries, but $I^2$ appears large because within-study variability was small, which is a natural consequence of large registry-based studies that benefit from large sample sizes [17, 57]. To accommodate potential heterogeneity, we conducted random effects meta-analysis.

In conclusion, our international study from four high-income countries indicates insufficient evidence for association between short IPI and adverse birth outcomes, except for a slightly elevated odds of spontaneous PTB for IPIs <6 months. We found consistently elevated odds of adverse birth outcomes for births following long IPIs. Previous recommendations for parents to wait at least 24 months may be unnecessarily long in high-income countries, while recommendations regarding long IPI are required.

## Supporting information

**S1 Checklist. RECORD checklist.**
(DOCX)

**S1 Fig. Flowchart of study inclusions and exclusions–Australia, Finland, Norway and California.**
(DOCX)

**S2 Fig. Adjusted odds ratios for the between-women analysis for the association between interpregnancy interval and preterm birth as compared to 18–23 months of**

**interpregnancy interval by country.**
(DOCX)

**S3 Fig. Adjusted odds ratios for the between-women analysis for the association between interpregnancy interval and spontaneous preterm birth as compared to 18–23 months of interpregnancy interval by country.**
(DOCX)

**S4 Fig. Adjusted odds ratios for the between-women analysis for the association between interpregnancy interval and small-for-gestational age birth as compared to 18–23 months of interpregnancy interval by country.**
(DOCX)

**S1 Table. Maternal characteristics at study cohort entry (first birth in the included cohort)**[*] **for the between-women analyses across the four countries (n = 3,849,193).**
(DOCX)

**S2 Table. Distributions of interpregnancy intervals by adverse birth outcomes in the cohort for the within-women analyses across the four countries.**
(DOCX)

**S3 Table. Sensitivity analysis–additional adjustment for socioeconomic status**[***] **in three countries.**
(DOCX)

**S4 Table. Sensitivity analysis–analysis excluding stillbirths in the interpregnancy interval calculation for within-women**[*] **analyses across the four countries.**
(DOCX)

**S5 Table. Sensitivity analysis–analysis considering women with ≥3 births/ ≥2 IPI cohort in the between-women analyses**[*] **across the four countries.**
(DOCX)

**S6 Table. Sensitivity analysis–analysis considering the first three births (parity 0, 1, and 2) in between-women**[*] **and within-women**[**] **analyses in Australia and Norway.**
(DOCX)

**S7 Table. Sensitivity analysis–analysis considering births from 1990 onwards in Norway in between-women**[*] **and within-women**[**] **analyses (1990–2016).**
(DOCX)

## Acknowledgments

The authors would like to thank the Linkage and Client Services Teams at the Data Linkage Branch (Department of Health Western Australia) as well as the Data Custodian for the Midwives Notification System. The authors would also like to thank members of the Healthy Pregnancies Consumer Reference Group who provided community input and guidance in this research.

## Author Contributions

**Conceptualization:** Gizachew A. Tessema, M. Luke Marinovich, Siri E. Håberg, Natasha Nassar, Stephen Ball, Amanuel T. Gebremedhin, Nick de Klerk, Maria C. Magnus, Cicely Marston, Annette K. Regan, Gary M. Shaw, Amy M. Padula, Gavin Pereira.

**Data curation:** Gizachew A. Tessema, M. Luke Marinovich, Mika Gissler, Jonathan A. Mayo, Maria C. Magnus, Gavin Pereira.

**Formal analysis:** Gizachew A. Tessema, M. Luke Marinovich, Mika Gissler, Jonathan A. Mayo.

**Funding acquisition:** Siri E. Håberg, Mika Gissler, Jonathan A. Mayo, Natasha Nassar, Stephen Ball, Gary M. Shaw, Amy M. Padula, Gavin Pereira.

**Methodology:** Gizachew A. Tessema, M. Luke Marinovich, Siri E. Håberg, Mika Gissler, Jonathan A. Mayo, Natasha Nassar, Stephen Ball, Ana Pilar Betrán, Amanuel T. Gebremedhin, Nick de Klerk, Maria C. Magnus, Cicely Marston, Annette K. Regan, Gary M. Shaw, Amy M. Padula, Gavin Pereira.

**Project administration:** Gizachew A. Tessema, Gavin Pereira.

**Resources:** Gavin Pereira.

**Software:** Gizachew A. Tessema, M. Luke Marinovich, Mika Gissler, Jonathan A. Mayo.

**Supervision:** Gavin Pereira.

**Validation:** Gizachew A. Tessema, M. Luke Marinovich, Mika Gissler, Jonathan A. Mayo, Natasha Nassar, Gavin Pereira.

**Writing – original draft:** Gizachew A. Tessema.

**Writing – review & editing:** Gizachew A. Tessema, M. Luke Marinovich, Siri E. Håberg, Mika Gissler, Jonathan A. Mayo, Natasha Nassar, Stephen Ball, Ana Pilar Betrán, Amanuel T. Gebremedhin, Nick de Klerk, Maria C. Magnus, Cicely Marston, Annette K. Regan, Gary M. Shaw, Amy M. Padula, Gavin Pereira.

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
