## [Decision Letter · Decision Letter 0]

16 Apr 2021

PONE-D-21-07581

Interpregnancy intervals and adverse birth outcomes in high-income countries: an international cohort study

PLOS ONE

Dear Dr. Tessema,

Thank you for submitting your manuscript to PLOS ONE. After careful consideration, we feel that it has merit but does not fully meet PLOS ONE’s publication criteria as it currently stands. Therefore, we invite you to submit a revised version of the manuscript that addresses the points raised during the review process.

We look forward to receiving your revised manuscript.

Kind regards,

Antonio Simone Laganà, M.D., Ph.D.

Academic Editor

PLOS ONE

Journal Requirements:

PLOS requires an ORCID iD for the corresponding author in Editorial Manager on papers submitted after December 6th, 2016. Please ensure that you have an ORCID iD and that it is validated in Editorial Manager. To do this, go to ‘Update my Information’ (in the upper left-hand corner of the main menu), and click on the Fetch/Validate link next to the ORCID field. This will take you to the ORCID site and allow you to create a new iD or authenticate a pre-existing iD in Editorial Manager. Please see the following video for instructions on linking an ORCID iD to your Editorial Manager account: https://www.youtube.com/watch?v=_xcclfuvtxQ

We note that you have indicated that data from this study are available upon request. PLOS only allows data to be available upon request if there are legal or ethical restrictions on sharing data publicly. For information on unacceptable data access restrictions, please see http://journals.plos.org/plosone/s/data-availability#loc-unacceptable-data-access-restrictions.

3a) If there are ethical or legal restrictions on sharing a de-identified data set, please explain them in detail (e.g., data contain potentially identifying or sensitive patient information) and who has imposed them (e.g., an ethics committee). Please also provide contact information for a data access committee, ethics committee, or other institutional body to which data requests may be sent.

3b) If there are no restrictions, please upload the minimal anonymized data set necessary to replicate your study findings as either Supporting Information files or to a stable, public repository and provide us with the relevant URLs, DOIs, or accession numbers. Please see http://www.bmj.com/content/340/bmj.c181.long for guidelines on how to de-identify and prepare clinical data for publication. For a list of acceptable repositories, please see http://journals.plos.org/plosone/s/data-availability#loc-recommended-repositories.

Additional Editor Comments :

The topic of the manuscript is interesting. Nevertheless, the reviewers raised several concerns: considering this point, I invite authors to perform the required major revisions.

Reviewers' comments:

Reviewer's Responses to Questions

**Comments to the Author**

1. Is the manuscript technically sound, and do the data support the conclusions?

Reviewer #1: Yes

Reviewer #2: Yes

Reviewer #3: Yes

Reviewer #4: Yes

2. Has the statistical analysis been performed appropriately and rigorously? 

Reviewer #1: Yes

Reviewer #2: Yes

Reviewer #3: Yes

Reviewer #4: Yes

3. Have the authors made all data underlying the findings in their manuscript fully available?

Reviewer #1: Yes

Reviewer #2: Yes

Reviewer #3: Yes

Reviewer #4: No

4. Is the manuscript presented in an intelligible fashion and written in standard English?

Reviewer #1: Yes

Reviewer #2: Yes

Reviewer #3: Yes

Reviewer #4: Yes

5. Review Comments to the Author

Reviewer #1: It might be worthwhile to include demographic characteristics stratified by IPI as well as country. Including raw OR results would also help clarify the effect of these demographic variables on this issue.

Socioeconomic status is discussed in the methods, but is not actually presented as an adjusted variable in the results. Further, since records of socioeconomic status vary by country, and do not provide a valid basis for comparison, you could consider dropping discussion and presentation of these variables altogether for a more efficient paper.

Reviewer #2: Dear Madam/Sir,

Re. PONE-D-21-07581_ORI

Thank you for inviting me to review this highly interesting and impressive manuscript (MS). As the current Senior scientist of one of the referenced papers (#19) and a citizen of one of the countries involved, it is a privilege to contribute. I also give beforehand credit to the 16 authors for their considerable time and efforts leading up to this MS.

Authors’ conclusions

In short, the overall and most clinically relevant conclusion is that a standard recommendation of 1½ to 2 years’ interpregnancy interval (IPI) may be open for revision towards a shortening, in particular for women in high income countries. That conclusion is based on a thorough follow up of national birth registry data covering more than 5 million pregnancies in four countries on three continents. By use of a robust design, meticulous and advanced data management and analyses, the authors make a very good case in support of their hypotheses.

Study outcomes and Gestational age

The study outcomes are a) Overall preterm birth (PTB), b) Spontaneous preterm birth (sPTB), and c) Small-for-Gestational Age (SGA) birth. All of them hinge on an ascertainment of gestational age that is as accurate as possible. The basis of the ‘exposure’ (IPI in months) is birth year of the mothers’ very first offspring and was collected over a period from 1980 to 2017. In most cases, Expected Date of Delivery (EDD) was ascertained by use of ultrasound (ULS). In case such information was unavailable, the peculiar (and hitherto novel) variable ‘expected date of conception’ was ascertained based on the woman’s self-reported menstruation history (LMP). Both EDD methods are probability predictions, but LMP is definitely more imprecise than ULS due to the normal variability of menstrual periods between 21 and 35 days. Moreover, routine ULS was no prenatal routine in Norway until well after a National Consensus conference in 1986.

Hence (1), I suggest a sensitivity analysis that compares births before 1985 and after 1990 to rule out any uncertainties as regards equivalent comparisons of the outcomes under study.

Maternal overall reproductive history

Nulliparae are exceptional since they, unlike parous women, have never given birth before. On the other hand, it is a frequent speculation that women tend to repeat the same outcome of consecutive births. In fact, the overall study hypothesis of Ref #19 (above) was to compare outcomes across sibships and therefore, selectively included women who expected their 2nd or 3rd child and whose earlier birth outcomes were known (Bakketeig et al., 1993).

Hence (2), I would like to see some clinical and other background information related to the ‘nullip’ pregnancies, i.e. in addition to the date when the delivery took place. This argument is relevant since any future modification of advice about IPI in a clinical setting normally must take that perspective into account. I have accessed the registration form for all births reported to the Norwegian Birth Registry, the most recent revision of which occurred in 1998. Three important variables that are recorded and should be considered are: a) Pre-eclampsia (varying from slight hypertension to full-blown eclampsia) is a strong predictor of SGA, b) PTB (overall) is recorded as one of the reasons for transfer of the child to the neonatal intensive care unit, and c) the mother’s current occupation (including branch) is recorded and listed as none, part-time or full time. The latter runs counter to the reported “N/A” in Table S1 (maternal occupation)

Maternal ethnicity

The overall part of the current study population is most likely Caucasian white, with US Californians as a potential exception. On the other hand, it seems to be a fact that for instance Afro-Americans have a slightly different and left-skewed gestational age profile. This may have an impact on all study outcomes that rely on predicted gestational length.

Hence (3), as a minimum I would like to see a comment about generalizability of study outcomes across ethnicity.

Extremely long IPIs and other reproductive aspects

I endorse the discussion related to the influence of IPIs of 60 months (= 5 years) and over. One aspect is the potential change of partner, a result of which is that the fetus is exposed to an entirely novel mix of maternal/paternal epigenetic imprint that in turn may influence the outcome. One example is the ‘return to null’ in terms of an increased risk for preeclampsia (PE; cf. Ref. 43) that add to the effect an advanced maternal age has on PE and other reproductive outcomes.

Hence (4), as PE is risk for SGA, a comment is requested about any potential effect of a consistent fatherhood or not.

Further, with an observed prevalence of about 1%, the authors claim that there is but a negligible influence on the outcomes under study in pregnancies after assisted reproduction. Their current comment is that the IPI has been inadvertently extended. However, recent data from the Norwegian Birth Registry indicates that close to one in ten pregnancies belongs to that category and should be taken into account in future analyses. On the other hand, this is a situation where the ‘date of conception’ mentioned earlier can be confirmed quite precisely.

Hence (5), please add or modify the current text. You may also consider to pay less attention to the cost issue?

Study design

PLoS ONE is a journal that aims broadly towards a general medical and other scientific readership. My impression and concerns after some 30 years are that the two words ‘retrospective’ and ‘cohort’ used together may (repeat: MAY) seem unfamiliar and contra-intuitive to a number of readers.

Hence (6), I ask the authors to present the study design as in the title and the abstract = ‘..an international longitudinal cohort study..’ In so doing, it underscores that these are the results of a standard follow up study of a single exposure (= ordinal levels of IPI) and three definite birth outcomes (= PTB, SPTB and SGA). The use of historic data - as shown by how they were collected - are in my view, obvious and sufficient.

In conclusion, this is an important and most valuable paper that merits publication in PLoS ONE. I enjoyed reading it and have already commended the authors for their initiative and the impressive amount of work they have laid down.

Data have been analysed according to state-of-art and the results presented precisely and diligently.

I hope my comments may be helpful in the upcoming process, whether they be taken into account or rebutted. My grading advice = Minor revision

Finally, I see no reason why these comments are kept secret.

Reviewer #3: In this manuscript the authors combine data from four countries to examine the effects of interpregnancy intervals on adverse outcomes, focusing on preterm birth, spontaneous preterm birth, and small for gestational age. The manuscript is presented in a straightforward manner, and in well-written scientific English. There are a few questions and concerns for the authors to consider.

First, with respect to definitions of IPI, the authors give readers a bit more detail. its a fairly complicated topic actually, on which knowledgeable researchers might disagree. Consider consulting one of two papers from an expert panel convened by the US Office of Population Affairs:

Hutcheon, Jennifer A., Susan Moskosky, Cande V. Ananth, Olga Basso, Peter A. Briss, Cynthia D. Ferré, Brittni N. Frederiksen, Sam Harper, Sonia Hernández-Díaz, Ashley H. Hirai, Russell S. Kirby, Mark A. Klebanoff, Laura Lindberg, Sunni L. Mumford, Heidi D. Nelson, Robert W. Platt, Lauren M. Rossen, Alison M. Stuebe, Marie E. Thoma, Catherine J. Vladutiu, Katherine A. Ahrens, “Good Practices for the Design, Analysis, and Interpretation of Observational Studies on Birth Spacing and Perinatal Health Outcomes”, Paediatric and Perinatal Epidemiology, 33,1 (January 2019), O15-O24. DOI: 10.1111/ppe.12512 PMID: 30311958

Ahrens, Katherine A., Jennifer A. Hutcheon, Cande V. Ananth, Olga Basso, Peter A. Briss, Cynthia D. Ferré, Brittni N. Frederiksen, Sam Harper, Sonia Hernández-Díaz, Ashley H. Hirai, Russell S. Kirby, Mark A. Klebanoff, Laura Lindberg, Sunni L. Mumford, Heidi D. Nelson, Robert W. Platt, Lauren M. Rossen, Alison M. Stuebe, Marie E. Thoma, Catherine J. Vladutiu, Susan Moskosky, “Report of the Office of Population Affairs’ Expert Work Group Meeting on Short Birth Spacing and Adverse Pregnancy Outcomes: Methodological Quality of Existing Studies and Future Directions for Research”, Paediatric and Perinatal Epidemiology, 33,1 (January 2019), O5-O14. DOI:10.1111/ppe.12504 PMID: 30300948

Either of these papers will provide readers with more context than is provided solely by reference 20 on p 5.

On p 6, first sentence of results, reading Table 1 it appears that the range of women at parity 1 outside the US was 89-93%, not 87-93% as stated. It would be helpful however to clarify how the term parity is used in the study. Are women of parity 1 giving birth to their first-born child, that is, nulliparous, or have they all had a previous live birth. This is unclear from the manuscript.

On a related point however, if we assume that parity=1 are women now giving birth to their second baby, it would be helpful to see the substantive analyses in Table 3 run also only on these births, as that would limit the analysis to birth intervals only for time from first birth to conception of the next pregnancy. As many other studies in the literature do this, more comparable results would then be available.

Also unclear is whether the analysis file links births into sibships, or if it is a strictly cross-sectional analysis across the years contributed for each country.

In conducting sensitivity analyses, it would be interesting also to control for race/ethnic variation. This probably can't be done in the pooled analysis, but should surely be possible in the US and Australia. Analyses including nativity would also be helpful.

Addressing these issues will result in a more broadly useful contribution to the international literature.

Reviewer #4: This manuscript presents some important findings. It uses large samples from 4 developed countries to examine IPA, using both between women and within women analyses. The later is important as it allows the analysis to adjust for unobserved selection bias. That this bias is present is confirmed by the fact that the effects are consistently smaller for the within women analyses. This analysis provides strong support that the current recommendation for a 24 month IPI is not supported for developed countries.

I have no concerns with the methods; they are solid and appropriate.

The results and concussions are clearly stated and the conclusions are supported by the results.

While the results as presented are fine, this reviewer found himself asking for some additional results that were not presented. Specifically,

1. Was very preterm considered as an outcome, in addition to preterm? The consequences of very preterm delivery are many times more serious than moderately preterm delivery. The authors have sufficient sample size to examine this, and it would be an important addition. That said, there is a lot of results in the current manuscript, and adding preterm may be beyond the scope of this study. If so, would encourage the authors to including it in an additional study.

2. The full regression results, including those used to generate the prognostic score, were not presented in the supplementary material. t is appropriate to not report these in the main text, as it would result in huge tables. But, the full results should be made available.

6. PLOS authors have the option to publish the peer review history of their article (what does this mean?). If published, this will include your full peer review and any attached files.

Reviewer #1: No

Reviewer #2: **Yes: **Geir W. Jacobsen, MD DrPH. Professor emeritus, Norwegian University of Science and Technology NTNU, Trondheim, Norway

Reviewer #3: **Yes: **Russell S. Kirby

Reviewer #4: No

---

## [Author Response · Author response to Decision Letter 0]

22 Jun 2021

Given that our responses included tables to address some of the comments by reviewers, we have provided our responses to reviewers as an attached file.

---

## [Decision Letter · Decision Letter 1]

8 Jul 2021

Interpregnancy intervals and adverse birth outcomes in high-income countries: an international cohort study

PONE-D-21-07581R1

Dear Dr. Tessema,

We’re pleased to inform you that your manuscript has been judged scientifically suitable for publication and will be formally accepted for publication once it meets all outstanding technical requirements.

Kind regards,

Antonio Simone Laganà, M.D., Ph.D.

Academic Editor

PLOS ONE

Additional Editor Comments (optional):

Authors performed the required corrections, which were positively evaluated by the reviewers. I am pleased to accept this paper for publication.

Reviewers' comments:

Reviewer's Responses to Questions

**Comments to the Author**

1. If the authors have adequately addressed your comments raised in a previous round of review and you feel that this manuscript is now acceptable for publication, you may indicate that here to bypass the “Comments to the Author” section, enter your conflict of interest statement in the “Confidential to Editor” section, and submit your "Accept" recommendation.

Reviewer #2: All comments have been addressed

Reviewer #3: All comments have been addressed

Reviewer #4: All comments have been addressed

2. Is the manuscript technically sound, and do the data support the conclusions?

Reviewer #2: Yes

Reviewer #3: Yes

Reviewer #4: (No Response)

3. Has the statistical analysis been performed appropriately and rigorously? 

Reviewer #2: Yes

Reviewer #3: Yes

Reviewer #4: Yes

4. Have the authors made all data underlying the findings in their manuscript fully available?

Reviewer #2: Yes

Reviewer #3: Yes

Reviewer #4: No

5. Is the manuscript presented in an intelligible fashion and written in standard English?

Reviewer #2: Yes

Reviewer #3: Yes

Reviewer #4: Yes

6. Review Comments to the Author

Reviewer #2: I commend (again) the authors for their intelligent responsiveness and their comprehensive efforts to accommodate my queries and demands

Reviewer #3: The authors have adequately addressed concerns of mine and the other reviewers. I have no other concerns at this time.

Reviewer #4: (No Response)

7. PLOS authors have the option to publish the peer review history of their article (what does this mean?). If published, this will include your full peer review and any attached files.

Reviewer #2: **Yes: **Geir Wenberg Jacobsen

Reviewer #3: **Yes: **Russell S. Kirby

Reviewer #4: No

---

## [Editor Report · Acceptance letter]

9 Jul 2021

PONE-D-21-07581R1 

Interpregnancy intervals and adverse birth outcomes in high-income countries: an international cohort study 

Dear Dr. Tessema:

I'm pleased to inform you that your manuscript has been deemed suitable for publication in PLOS ONE. Congratulations! Your manuscript is now with our production department. 

Kind regards, 

on behalf of

Dr. Antonio Simone Laganà 

Academic Editor

PLOS ONE